# A Structural Perspective on Calprotectin as a Ligand of Receptors Mediating Inflammation and Potential Drug Target

**DOI:** 10.3390/biom12040519

**Published:** 2022-03-30

**Authors:** Velia Garcia, Yasiru Randika Perera, Walter Jacob Chazin

**Affiliations:** 1Department of Chemistry, Vanderbilt University, Nashville, TN 37235, USA; velia.garcia@vanderbilt.edu; 2Center for Structural Biology, Vanderbilt University, Nashville, TN 37240, USA; yasiru.r.mahamarakkalage@vanderbilt.edu; 3Department of Biochemistry, Vanderbilt University, Nashville, TN 37240, USA

**Keywords:** S100 proteins, calprotectin, small molecule inhibitor, inflammation, RAGE, TLR4, CD33

## Abstract

Calprotectin, a heterodimer of S100A8 and S100A9 EF-hand calcium-binding proteins, is an integral part of the innate immune response. Calprotectin (CP) serves as a ligand for several pattern recognition cell surface receptors including the receptor for advanced glycation end products (RAGE), toll-like receptor 4 (TLR4), and cluster of differentiation 33 (CD33). The receptors initiate kinase signaling cascades that activate inflammation through the NF-kB pathway. Receptor activation by CP leads to upregulation of both receptor and ligand, a positive feedback loop associated with specific chronic inflammatory syndromes. Hence, CP and its two constituent homodimers have been viewed as potential targets to suppress certain chronic inflammation pathologies. A variety of inhibitors of CP and other S100 proteins have been investigated for more than 30 years, but no candidates have advanced significantly into clinical trials. Here, current knowledge of the interactions of CP with its receptors is reviewed along with recent progress towards the development of CP-directed chemotherapeutics.

## 1. Introduction

Calprotectin is a unique heterodimer of two S100 EF-hand calcium binding proteins, S100A8 and S100A9. CP functions in the host innate immune response to infection by pathogens, exploiting their need for transition metals to grow and thrive [1]. CP inhibits the growth of pathogens by chelating various transition metals with high affinity, effectively starving them of these essential nutrients, a mechanism termed nutritional immunity. CP also plays a role in the host inflammatory response by acting as a ligand for pattern recognition receptors including the receptor for advanced glycation end products (RAGE), toll-like receptor 4 (TLR4), and cluster of differentiation 33 (CD33) [2,3]. These receptors are activated by a wide range of ligands and signal through a MAPK-dependent kinase cascade that activates the NF-κB transcription factor. The resultant expression of inflammatory cytokines and chemokines, and generation of reactive oxygen species serves to stimulate inflammation. In addition, this signaling cascade leads to increased expression of ligand and receptor, creating a pro-inflammatory environment, i.e., as CP is secreted from immune cells it generates a positive, pro-inflammatory feedback loop (Figure 1) [4,5].

Inflammation is an essential physiological response to microbial invasion as well as cell stress and injury. However, aberrant regulation leading to chronic inflammation arises in a number of pathological conditions, which can result in tissue damage and disease progression. Inflammation can be triggered externally during infection by microbial pathogen-associated molecular pattern molecules (PAMPs) or internally as a response to cellular damage by damage-associated molecular pattern molecules (DAMPs) such as CP and other S100 proteins. Increased expression of DAMPs has been shown to correlate with Alzheimer’s, cancers, and autoimmune diseases (reviewed in [6,7,8]). Mis-regulation of the expression of S100 protein DAMPS has been implicated in several diseases and a number of S100 proteins are in use in the clinic as biomarkers, including for a range of cancers [9].

One of the best studied examples of CP disease association is inflammatory bowel disease (IBD), a chronic inflammatory condition characterized by prolonged inflammation of the gastrointestinal tract and resulting tissue damage [10]. Inflammation in IBD has been shown to be caused by an accumulation of DAMPs as well as an increase in the expression of pattern recognition receptors such as RAGE [11,12,13]. In fact, overexpression of CP is used as a biomarker for the diagnosis of IBD and can serve as a predictor for disease prognosis [14]. IBD is treated with non-specific anti-inflammatory drugs targeting cytokines such as TNFα [15]. However, these drugs are not effective for all patients and can lose efficacy over time. CP represents a potential target for the treatment of IBD and in this vein, other disorders mediated through CP-driven inflammatory signaling.

## 2. Biochemical and Structural Properties of CP

Calprotectin (CP) is a member of the S100 class of the EF-hand calcium-binding protein family. S100 proteins were first identified in 1965, named for their solubility in 100% ammonium sulfate [16]. Most genes encoding S100 proteins are found on chromosome 1q21 (this nomenclature arises from their position in the chromosome); over 20 members of the S100 class have been identified [17]. Among the EF-hand proteins, the S100 proteins are unique in that they are found almost exclusively in vertebrate animals, have a cell-type and tissue-specific distribution, and exhibit functions both inside and outside the cell [18,19].

EF-hand proteins are comprised of two or more helix–loop–helix motifs with a Ca^2+^ binding site centered in the loop, which together form a stable globular domain (Figure 2A,B). Canonical EF-hands have a specific array of 12 residues that chelate the Ca^2+^ ion using oxygen atoms from three side chains, one backbone carbonyl, one water molecule stabilized by a side chain, and both side chain oxygens of a strictly conserved glutamic acid in position 12 [20]. S100 proteins are distinguished by a non-canonical, N-terminal ‘S100-specific’ EF-hand, with the Ca^2+^ ion chelated through oxygen atoms arrayed over 14 residues including four from backbone carbonyls, one water molecule stabilized by a side chain, and both side chain oxygens of a strictly conserved glutamic acid in position 14. The EF-hand domains of S100 proteins are unstable on their own and instead form globular homodimers with their hydrophobic core integrated across the two subunits (Figure 2C) [21]. Although certain S100 proteins will form heterodimers, the S100A8–S100A9 heterodimer (CP) has a unique degree of complementarity between the two subunits that results in preferential formation of heterodimer over homodimer (Figure 2D) [22]. Like other S100 proteins, CP undergoes a conformational change as it binds Ca^2+^ that activates the protein for binding to its targets. In the case of CP, as the Ca^2+^ sites are filled, the protein also self-associates into a tetrameric dimer of CP heterodimers [23]. Oligomerization of S100 proteins has been shown to alter certain biochemical properties, e.g., tetramerization of CP protects the protein against bacterial proteases in vitro [24]. While interesting, there remains an urgent need to determine the larger physiological and pathologic relevance of oligomerization of CP and other S100 proteins.

S100 proteins also bind zinc and other transition metals at two sites at the dimer interface (Figure 2C,D). The CP heterodimer is distinct from other S100 proteins in that it contains two different transition metal binding sites [25,26]. One site is a canonical His_3_Asp site that binds zinc and copper with high affinity. The second site chelates ions with six His side chains, which is unique among all S100 proteins because two His residues are provided by a C-terminal His-rich extension found only in S100A9. In addition to zinc and copper, this non-canonical site also binds manganese and ferrous iron [25,27]. Although transition metals bind at a distinct site from calcium, they have also been shown to promote formation of tetramers and higher order oligomers in vitro [28].

## 3. CP as a Ligand of Inflammatory Receptors

As noted above, CP activates multiple inflammatory receptors including RAGE, TLR4, and CD33. In this section, we summarize knowledge of the structural basis for activation of these three receptors and what is known about CP as a ligand. Physical and functional interaction of CP with other receptors have been reported, although whether they are involved in inflammation is not well defined [29,30]. Additional uncertainty arises because in many of these studies, the S100A8 and S100A9 homodimers are not distinguished from the CP heterodimer. Careful consideration of this potential complication is relevant in assessing the ability to target CP-meditated inflammatory signaling.

### 3.1. RAGE

The receptor for advanced glycation end-products (RAGE) is a multi-ligand pattern recognition receptor that mediates a range of inflammatory processes and its ligands have been implicated in several inflammatory conditions and inflammation-driven pathologies [31]. As noted above, RAGE is activated upon interaction with DAMPs and meditates inflammation through the NF-ĸB signaling axis. RAGE is associated with certain chronic inflammatory disorders as well as Alzheimer’s Disease and diabetes [32]. RAGE is present at low concentrations in vascular tissue, but its expression is upregulated in the presence of its ligands. Overexpression of RAGE creates a pro-inflammatory environment, which also leads to the overexpression of more RAGE ligands creating a positive feedback loop [33].

RAGE is composed of three extracellular Ig-domains (V, C1, C2), a single membrane spanning domain, and a small 43 residue C-terminal cytosolic tail. The RAGE V and C1 domains have been shown to form an integrated structural unit (VC1) that is essential for high affinity ligand binding [34]. RAGE binds a wide variety of ligands, including non-enzymatically glycated adducts termed “advanced glycation end products” (AGEs) from which its name is derived. The first native ligand of RAGE to be reported was S100A12, and RAGE has been found to serve as a receptor for a number of S100 proteins including CP [35].

The structural mechanism of RAGE activation has been investigated for several S100 proteins. Binding to RAGE is triggered by the calcium-induced conformational change that exposes a shallow hydrophobic surface within each subunit of the S100 dimer. Solution NMR studies mapping the RAGE-S100B [36] and RAGE-S100A12 [37] binding interfaces reveal that in addition to the interaction having a hydrophobic component, complementarity between the basic surface of the VC1 domain and the acidic target binding surface of S100 proteins also play a role. An NMR structure [38] was reported of S100A6 bound to RAGE V, whereas a 2.4 Å crystal structure of S100A6 bound to the VC1 domain of RAGE shows S100A6 interacting primarily with the C1 domain of RAGE [39]. Moreover, binding of S100A6 to RAGE C2 by surface plasmon resonance has also been reported [40]. Physical and functional interaction with all three RAGE extracellular domains is unlikely and further, more definitive analysis is required to resolve this controversy.

Ligand-induced oligomerization is a common activation mechanism for Ig-like receptors like RAGE, and indeed RAGE signaling is mediated by trans-phosphorylation of kinases recruited to the cytosolic surface of the membrane by the small intracellular domain (reviewed in [41]). Multiple phosphorylation events are needed to generate a sufficiently strong signal downstream, so all current models have receptor aggregation as a common core element. Although zinc-mediated dimerization of RAGE has been reported [36], oligomerization of RAGE ligands is generally believed to be critical to assembling a sufficient number of RAGE molecules to initiate signaling downstream. Although not characterized in a direct manner, this view is supported by a variety of indirect studies, including a report that RAGE has a higher affinity for S100B in an unusual tetrameric state than the more prevalent dimer [42]. CP, which exhibits multiple metal-induced oligomerization states, is expected to be particularly effective at activating RAGE.

### 3.2. TLR4

Toll-like receptors (TLRs) are another class of pattern recognition receptor in innate immunity. They are widely expressed in both immune (dendritic cells, monocytes, mast cells, and macrophages) and non-immune (fibroblasts, epithelial cells) cells (reviewed in [43]). TLRs have important roles in host defense against pathogenic organisms and are evolutionarily conserved throughout mammals [44]. Ten TLRs have been identified in humans. They are activated by a variety of ligands including DAMPs and PAMPs, which, like RAGE, lead to NF-ĸB signaling and production of inflammatory cytokines. However, the mechanism of activation of TLRs is more complex than RAGE as their activation requires co-receptors including MyD88 and TRIF [45,46]. In most cases, the initial binding of ligands stimulates the TLR to interact with the co-receptor and it is the complex that is then able to initiate the signaling cascade.

TLR4, like all TLRs, is composed of three domains: an extracellular domain, a transmembrane domain [47], and an intracellular Toll-interleukin-1 receptor (TIR) domain (reviewed in [48]). Three-dimensional structures of TLR4 are available but limited to the individual domains. The 608 residue extracellular domain is a network of leucine-rich repeats (LRRs) arranged in a horseshoe-like shape with distinct concave and convex surfaces [48]. The short transmembrane domain of TLR4 forms a helical structure of 21 residues followed by a loop while the 187 residue intracellular region forms a canonical TIR domain [49,50]. TLR4 downstream signaling is driven by the binding of its TIR domains to the intracellular mediators of the cascade such as type I interferon (IFN) and inflammatory cytokines.

TLR4 is activated by PAMPs such as lipopolysaccharides (LPS) and DAMPS including S100 proteins through the co-receptors MD2 and CD14 (reviewed in [51]). Activation by LPS is the most well studied. CD14 binds LPS and loads it onto the complex of TLR4 and MD-2 (TLR4–MD-2) [45]. The transfer of the LPS substrate induces a conformational change in TLR4 that, in turn, leads to the formation of the heterodimer of TLR4 and MD-2, which, in turn, triggers the NF-κB signaling pathway. The activation of TLR4 by S100 proteins is often compared to the activation by LPS. For example, like LPS, S100 proteins bind tightly to TLR4. However, although CD14 is involved, evidence has accumulated suggesting that S100 proteins directly interact with TLR4 and do not require MD-2 for binding [52], which implies they activate signaling by a distinct mechanism. The affinity of TLR4–MD-2 for the S100A12 complex has been reported to be in the nanomolar range [53]. Binding requires calcium or zinc but the highest affinity was measured when both are bound. In contrast, using an ELISA assay, only modest affinity was found for the binding of the S100A8 homodimer to the TLR4–MD-2 complex and MD-2 alone (0.32 and 0.73 µM, respectively) [52].

CP activation of TLR4 signaling has been studied by Vogl et al. [54]. They reported that TLR4–MD-2 signaling can be induced by S100A8 and S100A9 homodimers and the CP heterodimer, but not the CP tetrameric state. A model was presented in which CP dimer secreted from cells will activate TLR4–MD-2 before the tetrameric state is induced by the high Ca^2+^ concentration in the extracellular milieu. However, once the tetramer is formed, it inhibits activation of the complex. In further support of this model, they report indirect evidence suggesting that TLR4–MD-2 binds the canonical C-terminal EF-hand of S100A9 in the CP dimer, but the same interaction interface will be blocked in the CP heterotetramer. Interestingly, the previously noted study of S100A8 dimer also proposed that binding to TLR4–MD-2 was mediated by the canonical C-terminal EF-hand [52].

### 3.3. CD33

CP-mediated activation of the pattern recognition receptor CD33 represents a relatively new direction for investigation of the role of CP in inflammation and is much less characterized than for RAGE and TLR4. CD33 is a member of the Siglec (sialic acid-binding Ig-like lectins) receptor family, implicated in certain cancers and Alzheimer’s disease [55]. It consists of an extracellular region with two Ig-like V-type domains and an Ig membrane (C2-set) domain, a membrane spanning region, and an intracellular domain with an immunoreceptor tyrosine-based activation motif (ITAM) that serves to mediate signaling. CD33 is expressed in humans as a 67 kDa transmembrane glycoprotein that preferentially binds to alpha-2,6-linked sialic acid [55]. The sialic acid binding site of CD33 contains a conserved arginine residue that is positively charged at physiological pH [56]. This positively charged arginine residue is assumed to have the prototypical function of forming a salt bridge with salicylic residues conjugated to the glycoprotein.

In addition to salicylic acids, S100 proteins are also recognized as CD33 ligands [57]. In particular, inflammation associated with myeloid-derived suppressor cell (MDSC) expansion was shown to be promoted by S100A9 interaction with CD33 [3]. Further studies revealed the involvement of CP in MDSC expansion and hematopoietic progenitor cell (HPC) apoptosis [58]. In one report, the plasma concentration of S100A9 was found to be significantly increased in myelodysplastic syndrome (MDS) patients [59]. In a murine model, S100A9 drove expansion and activation of MDSCs, which contributed to cytopenia and myelodysplasia [60]. Despite there being strong evidence of S100 protein binding and activation of CD33, little is known about the roles of S100A8, S100A9, and CP metal-dependency, oligomerization, structure, and dynamics. In particular, there is a dearth of careful study to distinguish between S100A9 homodimers and the CP heterodimer. Overall, there is much to be learned about the activation of CD33 by CP and other S100 proteins, and the potential therapeutic value of targeting their interaction.

## 4. CP as a Drug Target

Given the many connections of CP and other S100 proteins to chronic inflammation and specific diseases, there has been sustained interest in discovering, testing, and refining small molecule inhibitors for better understanding biochemical pathways and evaluating the potential therapeutic value of these molecules [59]. A variety of different approaches have been used to identify S100 protein inhibitors from direct small molecule screening to structure-based molecular design. In the following, we first provide an overview of progress organized by the various strategies used for small molecule discovery of S100 protein inhibitors, since these are all potential strategies that could be applied to CP. We then describe the current status of CP drug discovery and end with a brief overview of complementary progress to develop drugs that are targeted to the inflammatory receptors for which CP is a ligand.

### 4.1. Inhibitors of S100 Proteins

The first S100 protein inhibitors were found serendipitously in very broad, general searches to identify the target(s) of drugs whose mode of action was not firmly established at the time. One of the earliest studies involved the anti-allergy drugs amlexanox, cromolyn, and tranilast, which were known to inhibit IgE-meditated degranulation of mast cells [61]. Affinity chromatography experiments using bovine lung lysate identified a number of S100 proteins as binding partners. The interaction of specific S100 proteins with these molecules was then investigated. Binding of tranilast to S100A12 was shown to block the interaction with the RAGE V domain [37]. Similarly, binding of cromolyn to S100P was shown to block interaction with RAGE [62]. Analogs of cromolyn have been developed that show enhanced binding to S100P, which was proposed to minimize its off-target effects [63]. However, this hypothesis has yet to be tested. Although these molecules show chemotherapeutic effects in cells, they are not specific for any one S100 protein and have been shown to interact with a number of kinases as well [64,65]. Similarly, the phenothiazine class of antipsychotic drugs were shown using affinity chromatography to bind to S100B and S100A1 as well as other EF-hand calcium-binding proteins [64]. Although these molecules bind a number of EF-hand proteins, structural studies have revealed that the binding modes for each protein are quite different [66,67,68]. Efforts to improve specificity of these molecules are ongoing [64,69,70].

A number of S100 protein drug development efforts have utilized high throughput screening (HTS) combined with structure-based molecular design. A fluorescence polarization competition assay (FPCA) was developed for HTS of S100B to selectively screen small libraries of compounds that can bind Ca^2+^-bound S100B and inhibit its interaction with a p53 peptide in vitro [68]. The assay was based on the peptide termed TRTK-12, which is known to bind at the S100B target interface. The peptide was labeled with a fluorophore and changes in fluorescence anisotropy were used to monitor release from S100B when it was outcompeted by a ‘hit’. In another campaign, an inhibitor screen of FDA approved drugs was carried out for S100A4 using a fluorescent biosensor and protein conjugate, Mero-S100A4 [71]. The Mero biosensor is linked covalently to S100A4 via Cys81 and binds along the target binding interface of S100A4 with two molecules of Mero bound per S100A4 dimer. This screen identified a number of molecules that bind S100A4, including phenothiazines. An X-ray crystal structure of phenothiazine trifluoperazine (TFP) bound to S100A4 revealed two TFP molecules bound in the hydrophobic cleft of S100A4 with four molecules bound per dimer [72]. This 4:1 ratio was sufficient to induce oligomerization of S100A4 to a pentamer. Interestingly, inhibition of the S100A4 function was observed only when the concentration of TFP was high enough to induce oligomerization [72].

Virtual screening has been used as a starting point for some S100 protein drug discovery programs. After screening the large number of potential hit molecules in silico, hits are verified by biophysical methods such as NMR. Verification of hits by NMR for S100B enabled prioritization of those interacting at the target binding interface. The structure of S100B in complex with one such hit, SEN205A, was then determined by X-ray crystallography [73]. In another study, inhibitors of S100A10 were discovered using virtual screening and subsequent analysis by a competitive fluorescence binding assay [74]. After the optimization of hits based on structural data, analogs of 4-aroyl-3-hydroxy-5-phenyl-1*H*-pyrrol-2(5*H*)-one and substituted 1,2,4-triazoles were synthesized and then shown to inhibit the interaction of S100A10 with an annexin A2 peptide [74].

After screening to find hit molecules, structure-guided elaboration is greatly facilitated by knowledge of target binding sites, which have been analyzed for their unique chemical and structural characteristics [75]. Moreover, there are numerous structures of S100 proteins bound to peptide fragments of their targets. Among these, the interaction of S100B and its target p53 is one of the most well characterized (reviewed in [76]). This interaction is calcium-dependent; S100B, like other S100 proteins, undergoes a conformational change upon calcium-binding that exposes a hydrophobic patch that is essential for binding to p53. A virtual screen using high resolution crystal structures of S100B and the S100B–p53 complex was used to screen a library of compounds to predict inhibitors [72]. Seven of the virtual hit molecules were found to inhibit the S100B interaction with a p53 peptide, one of which was the FDA approved drug pentamidine. Chemical elaboration of pentamidine using a structure-guided approach resulted in tighter binding to the protein [69].

The target-binding site of S100B has also been characterized by structures of complexes with small molecule inhibitors (Figure 3). Interestingly, these bind in a variety of different modes. The hydrophobic patch of S100B has been subcategorized into three distinct pockets, termed Site 1, Site 2, and Site 3 [69]. Site 1 is the p53 target binding site where p53 fragments interact with Helices 3 and 4, as well as a peptide derived from RAGE (Figure 3B,C). The above noted small molecule SEN205A binds in this Site 1 region, inhibiting by directly blocking the p53 binding site. Surprisingly, the majority of S100B inhibitors such as pentamidine and heptamidine bind at Site 2 and Site 3, interacting with Helix 4 and the loop region on Helix 2 (Figure 3D,E). As these inhibitors are able to block p53 binding without binding directly in the target binding site, they are presumably inducing allosteric structural effects that disrupt target binding. A structure-based approach has been used to further optimize these compounds, and a number of these S100B-inhibitor complexes have been structurally characterized [70]. All ‘hit’ compounds were tested for inhibition in the presence and absence of calcium to make sure that they are inhibiting the relevant form of the protein [70]. Binding to S100A1 was also investigated to begin to test the specificity of these molecules among the S100 proteins.

### 4.2. Inhibitors of Calprotectin

The large majority of drug discovery efforts for CP derive from an ongoing program at the company Active Biotech (Lund, Sweden). This effort originated with a broad screen to identify novel anti-inflammatory molecules, which turned up a number of quinoline-3-carboxamides (Q compounds) with interesting anti-cancer and immunomodulatory properties [77]. The focus of the program was initially on roquinimex, but it and other analogs were found to elicit pro-inflammatory effects [77]. In order to reduce the pro-inflammatory properties of these molecules, a number of second-generation Q-compounds were synthesized and tested for anti-inflammatory activity in animal models for multiple sclerosis. This resulted in a number of newer lead molecules, including laquinimod (ABR-215062), tasquinimod (ABR-215050), and paquinimod (ABR-215757) [78] (Figure 4). These three molecules are currently in Phase 2 and 3 clinical trials for the treatment of various cancers and inflammatory disorders such as multiple myeloma, prostate cancer, and uveitis. Five years ago, Active Biotech released for study ABR-238901, which is purported to be an inhibitor of both S100A9 homodimer and CP. This molecule has been used in multiple studies in cells and mouse models of inflammatory diseases, revealing that treatment with ABR-238901 dampens inflammation [79,80,81]. However, no data have been reported that directly confirm this molecule binds with significant affinity to S100A9 homodimers or CP, or that in cells, this molecule binds only to S100 proteins.

The potential therapeutic value of these S100A9-binding Q compounds has been investigated in several models. It has been proposed that Q-compounds inhibit tumor growth by blocking S100A9 activation of TLR4-mediated inflammation [82]. In support of this hypothesis, treatment with tasqinimod has been shown to inhibit tumor growth in a mouse model of prostate cancer in a manner that mimics the phenotype seen in S100A9 deficient mice [83]. In addition, S100A9 is known to be secreted from myeloid-derived suppressor cells (MDSC) and serves as a chemoattractant that promotes MDSC recruitment and tumor signaling pathways. Therefore, the anti-carcinogenic effects of S100A9-binding compounds in this context appear to arise from inhibition of S100A9-dependent recruitment of MDSCs [82].

In an investigation into the mechanism of action of Q-compounds the authors concluded that they meditate inflammation by directly binding to S100A9 [84]. In that study, biophysical analysis in vitro showed that Q-compounds interact specifically with S100A9 homodimers and not CP or S100A8 homodimers. In addition, binding of Q-compounds to S100A9 assayed by surface plasmon resonance was reported to block binding to RAGE and TLR4. Despite evidence to the contrary in the literature, the authors proposed that only the S100A9 homodimer and not CP bind to RAGE and TLR4 [84]; further investigation will be required to further test the generality of this hypothesis.

### 4.3. Receptor-Targeted Drugs

Because of its role in inflammation, there has been significant interest in RAGE as a therapeutic target [85]. RAGE-binding peptides have been designed from the structure of one of its ligands, S100P [86]. In addition, the small molecules including Azeliragon and FPS-ZM1 [87] have been shown to inhibit the activity of RAGE [88]. In vitro biochemical data show that these molecules are able to inhibit RAGE activation by different binding partners [87,89]. However, there are no structural data that would define the mechanism of inhibition for any of these molecules. Despite the intense interest and existence of RAGE inhibitors molecules, with some in clinical trials, there are no FDA approved drugs for RAGE [90].

TLR4-targeted drugs are also of intense interest [91] in the context of a variety of autoimmune diseases [92], infection-induced sepsis [93], and inflammatory bowel disease [94], all associated with S100 proteins. A number of small molecule inhibitors have been developed to inhibit TLR-mediated inflammation pathways. TAK-242 is a small molecule inhibitor that has been shown to bind to TLR-4 and block interactions with its adaptor molecules [95]. Although this inhibitor was shown to be specific for TLR-4, it did not make it past Phase 3 clinical trials as it was not able to significantly suppress serum cytokine levels in septic patients compared to controls. Other small molecule inhibitors such as Corilagin [96] and Artesunate [97] have been shown to block LPS-induced inflammation in a TLR-4 dependent manner. However, here again, the absence of structural information has limited the understanding of the mechanism of action of these drugs.

CD33 is also an attractive therapeutic target for acute myeloid leukemia (AML) and neurodegenerative disorders. There have been clinical trials on drug–antibody conjugates that target CD33 [98]. In addition, targeting CD33 with sialic acid mimetics has been successful in inhibiting CD33 in vitro [99]. There are, however, no reports of drug-like molecules that bind CD33.

## 5. Discussion

CP has been identified as an important biomarker in inflammatory diseases, but many challenges have slowed understanding what roles CP plays in different cellular and tissue contexts. (1) S100A8 knock-out mice are not viable. (2) The only available inhibitors appear to be specific for S100A9 homodimers, but none for CP or S100A8 homodimers. (3) There is a lack of comprehensive understanding of the structure and dynamics of S100 protein–receptor interactions and no high-resolution structures of S100 proteins in complex with an intact, full-length receptor to inform mechanisms of signaling. (4) Current evidence indicates that there are multiple modes of S100 proteins binding to receptors and of small molecule ligands to S100 proteins. (5) Oligomerization and post-translational modifications of S100 proteins are known to affect receptor interaction but are poorly characterized. The availability of CP-specific inhibitors would represent a major step forward for furthering the understanding of CP-mediated inflammatory signaling and association with chronic inflammatory syndromes.

The availability of S100 protein inhibitors has helped elucidate the roles that CP as well as other S100 proteins play in inflammation and clarify their association with chronic inflammatory disorders. Although high-throughput screening in combination with structure-based design have produced a number of S100 protein inhibitors, there are no FDA-approved drugs specific for any S100 protein to date. Specificity among different S100 proteins poses a significant challenge, as the molecules identified in screens for one S100 protein often bind to other members of the family. CP may be a uniquely challenging case in that the target binding sites of both S100A8 and S100A9 homodimers are expected to be similar to those in the heterodimer. Although the physiological importance of the tetrameric state of CP remains uncertain, the facile calcium and transition metal-induced tetramerization of the CP is a distinctive feature and may offer unique opportunities for generating specificity of inhibitors.

The large width of the target binding sites on S100 proteins points to the potential for applying the fragment-based drug discovery approach to CP inhibitor development. The ability to map the S100B protein target interface provides further support for pursuing this approach. The screening of fragments (<300 Da) as opposed to larger compounds would allow for the identification of multiple sub-pockets across the wide-target binding interface that can be linked together to generate the requisite high overall affinity. This approach has the potential to ‘read’ the fine contours of the S100 protein binding site being probed, which could greatly aid in generating S100 protein specificity. We anticipate this approach being implemented in future CP-directed drug discovery campaigns.

## Figures and Tables

**Figure 1 biomolecules-12-00519-f001:**
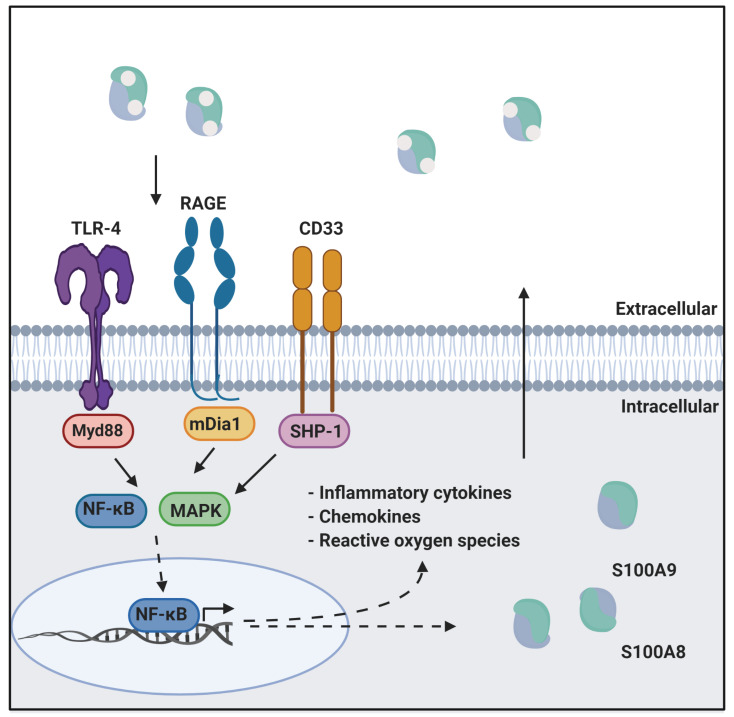
CP in innate immunity. CP serves as a ligand for inflammatory receptors such as RAGE and TLR4. Activation of these receptors through the MAPK dependent signaling cascade activates the NF-κB transcription factor. This results in the expression of cytokines, chemokines, and reactive oxygen species that drive inflammation. This figure was created using BioRender.com (last accessed on 10 February 2022).

**Figure 2 biomolecules-12-00519-f002:**
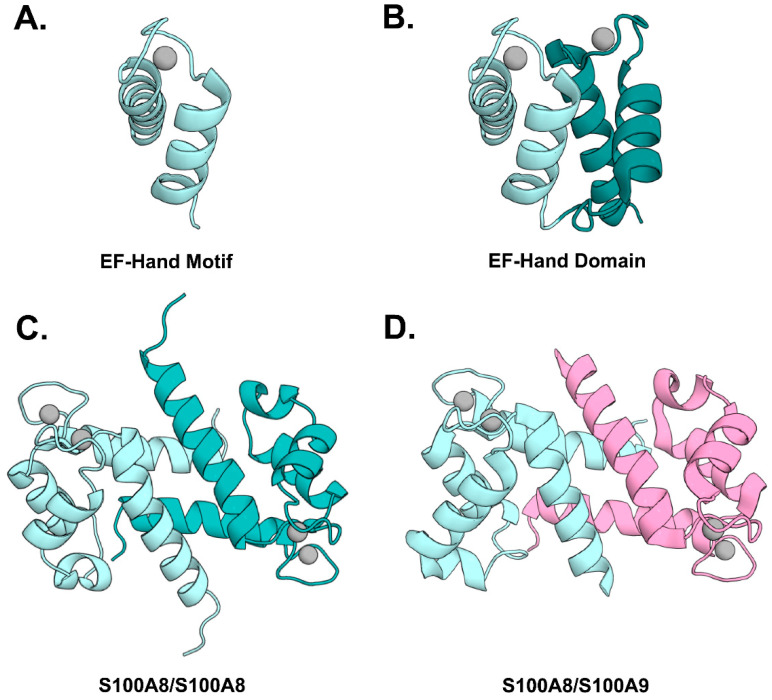
Structural features of S100 proteins: Upper left panel shows a helix–turn–helix EF-hand Ca^2+^-binding motif (**A**) and an EF-hand domain (**B**) (PDB entry 2H61). The lower panel shows the integration of two EF-hand domains in the S100A8 homodimer (**C**) (PDB entry 1MR8) and the S100A8–S100A9 heterodimer (**D**) (PDB 1XK4).

**Figure 3 biomolecules-12-00519-f003:**
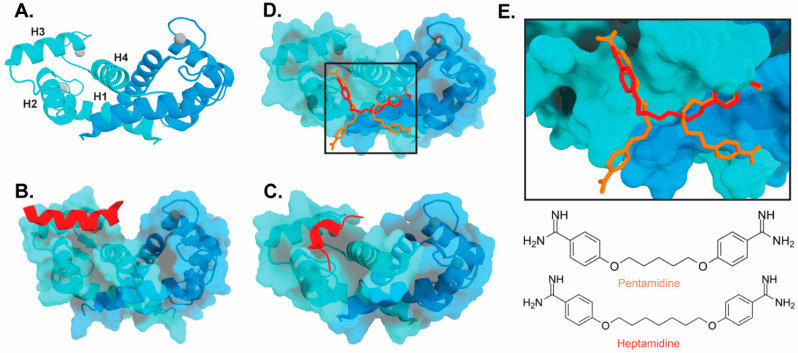
Binding of ligands to S100B: (**A**) Ribbon structure of calcium-bound S100B dimer (PDB 3D10) individual subunits are marked in blue and cyan with calcium represented as gray spheres. The four helices of one S100B subunit are labeled H1–H4. (**B**) S100B bound to p53-derived peptide (PDB 1DT7). (**C**) S100B bound to RAGE-derived peptide W61 (4XYN). (**D**) S100B bound to two molecules of pentamidine and one molecule of heptamidine (4FQO). (**E**) Zoomed in view of panel D binding site. Chemical structures are shown below.

**Figure 4 biomolecules-12-00519-f004:**
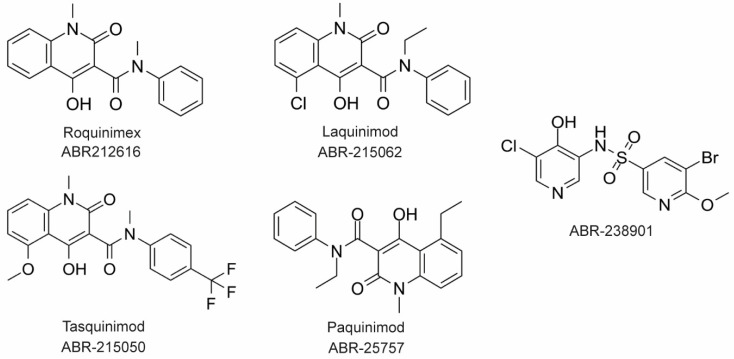
Chemical structures of Active Biotech CP inhibitors reported to date.

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
