# Peer review of "A Structural Perspective on Calprotectin as a Ligand of Receptors Mediating Inflammation and Potential Drug Target"

_biomolecules, 2022, doi:10.3390/biom12040519_

Round 1

Reviewer 1 Report

The manuscript written by Chazin et al. describes the interaction of calprotectin with its receptors and potential ways of using chemotherapeutic agents in this pathway. In my opinion, the work is written in a general way, the authors do not go deeply into the mechanisms or the structural approach.

Below, there are some exemplars, which in my opinion would help to improve the quality of the manuscript.

Figure 2 – missing A, B, C, D notations; in my opinion, this figure should exhibit important amino acid residues engaged.

Figure 3 – what H means? I suppose helix, but in my opinion, it should be explained.

The manuscript concerns the possible use of chemotherapeutic agents, but mechanisms and details about the exemplary compounds are missing (eg. their structures, the way in which they interact)

Abbreviations should be explained the first time they are used, later authors should use abbreviations.

Literature cited by the Authors relates rather to earlier years.

Reviewer 2 Report

In this manuscript, Garcia et al. review the role of the S100A8/A9 heterodimer, also known as calprotectin, in promoting inflammation through interaction with cell surface receptors. They further discuss how this may contribute to chronic inflammation in different pathological settings and how this role poses calprotectin and its receptors as valuable drug targets for anti-inflammatory applications. In a second part, the authors detail the various drug design strategies that have been developed to select small molecule inhibitors against S100 proteins in general, using structural information available on S100 members and their complexes with effector molecules. Finally they highlight how these general principles have been applied to conceive inhibitors targeting calprotectin and its pro-inflammatory receptors.

The review is well in line with the Special Issue on S100 proteins to which it is submitted and it offers interesting and novel perspectives in that 1) it provides a structural point of view on the pro-inflammatory role of calprotectin and 2) it gives a rather complete overview of the various small molecule inhibitors of S100 proteins that have been developed so far. The manuscript is clear and well written. The review would however gain in depth by addressing some points in more details and illustrate some of the data more clearly. Furthermore, an entire paragraph in the manuscript is missing due to duplication of the previous paragraph. This missing part should be provided to the reviewers before any decision can be taken. I have listed below comments that should be addressed by the authors:

1) Paragraph 4.3 on Receptor-targeted drugs is missing. The current text in the manuscript is a duplicate of paragraph 4.2. This should be corrected and the new version of the manuscript should be sent back to the reviewers for reading as this is an important chapter of the review.

2) In the Introduction, the authors mention that calprotectin interacts with various cell-surface receptors including RAGE, TLR4 and CD33. An entire part of the review is then dedicated to the molecular details of these interactions.

2a) RAGE and TLR4 have been extensively described as receptors for S100A8/A9. On the other hand, data on CD33 and S100A8/A9 are scarcer and, to my knowledge, only mention an interaction with S100A9 rather than with calprotectin. Although the S100A8/A9 heterodimer seems to be predominant in vivo as compared to the respective A8 and A9 homodimer, one cannot rule out that CD33 only interacts with the S100A9 homodimer. This should be clearly stated in the manuscript.

2b) Other receptors for the S100A8/A9 heterodimer and/or its individual components have been described in the literature, including CD147 (Basigin), melanoma cell adhesion molecule (MCAM) and neuroplastin β. Although these interactions are not directly pro-inflammatory and rather seem to contribute to cancer cell migration and metastasis, they do highlight the complexity of the calprotectin interactome. Other reports have even suggested that RAGE or TLR4 may not be the major vehicles of S100A8/A9 pro-inflammatory signal in vivo (e.g. Chen et al 2015 PLoS One). While many of these interactions are still speculative, the review should acknowledge the complexity and the variety of calprotectin-receptors interactions. For example, line 108 states that calprotectin binds to three pro-inflammatory receptors (no more no less). This should be rephrased to “CP activates at least three different inflammatory receptors”. Similarly, a short conclusion on this part could be added to emphasize that other receptors yet unraveled may exist for the transduction of CP pro-inflammatory signals, highlighting how this may complicate drug design approaches to target CP-mediated signaling.

2c) In Figure 1, the available details on CD33 downstream signaling following S100A9 challenge should be indicated (e.g. SHP-1 recruitment), at least with a question mark. Does this receptor also lead to NFKB activation? Lines 32-33 of the introduction, it is stated that CD33, as RAGE and TLR4, signals to activate MAPK-pathways and NFKB. If true, this should be added in Figure 1. If unknown yet, the paragraph in the introduction should be corrected.

3) Lines 53 to 62: the authors are confusing irritable bowel syndrome (IBS) and inflammatory bowel diseases (IBD). Calprotectin serves as an inflammatory marker in both diseases. The authors should clarify which disease they refer to. Although wrongly assumed for a long time, it is now well-established that IBD is not an autoimmune disease and that it has a much more complex etiology (genetic deficiencies + environmental factors + dysbiotic microbiota), this should also be corrected.

4) Lines 55-57: the reference cited to highlight the role of RAGE and DAMPs accumulation in IBD is not proper (ref. 11). The authors should rather cite more appropriate literature, e.g. Body-Malapel et al. 2019 Mucosal Immunology for the role of RAGE signaling in IBD or Nanini et al. 2018 World J Gastroenterol for the role of DAMPs in IBD.

5) Line 113: “RAGE is a multi-ligand receptor that mediates certain inflammatory processes”. I find this sentence a bit reductive as RAGE and its ligands have been implicated in many inflammatory conditions and inflammation-driven pathologies. This should be rephrased.

6) The paragraph on inhibitors of S100 proteins (Paragraph 4.1) is a bit difficult to follow especially when it comes to structural consideration. More illustrations on this part may ease the reading. In particular, when available, structures of complexes between selected S100 and their inhibitors may be displayed in a zoomed view to highlight the different modes of actions of these drugs (through blockade of different S100 epitopes with distinct functions). This is partially attempted in Figure 3 but the figure is rather small and sticks to views of the entire S100 dimer. The different panels could easily be combined in a figure more zoomed on the inhibitor binding pocket so that the pentamidine molecules are more clearly visualized. Other panels with other S100-inhibitor structures could then be added.

7) Lines 412-413: “To date there are no high-resolution structures of S100 proteins in complex with an intact receptor”. I do not agree with this statement. There are some structural data to reasonably high resolution for complexes with entire ectodomains of the receptor(s). In the case of RAGE for example, it will be highly difficult to obtain a structure of the full-length receptor as this is a single-pass transmembrane receptor. The major problem is rather that there are too many structural data, all contradictory. I would suggest to modulate this statement so that it fits better with the available data in the literature.

Round 2

Reviewer 2 Report

The manuscript by Garcia et al. has now substantially improved. All my concerns/questions have been answered and modifications of the figures as requested is fine. I would therefore recommend this manuscript for publication in Biomolecules providing one last minor correction is added:

1) Lines 55-56: again, IBD is for Inflammatory Bowel Diseases and not Irritable Bowel Disease (the correct term is Irritable Bowel Syndrome, abbreviated IBS). IBD corresponds to ulcerative colitis and Crohn's disease, 2 pathologies that are medically distinct from irritable bowel syndrome. The correct terminology should be used here. From the rest of the paragraph, I assume the authors talk about IBD (which is indeed treated by anti-TNFalpha) so they should replace "irritable" by "inflammatory" line 55.

Author Response

We thank the reviewer and have fixed this in the text.